# Do Not Withhold Mitral Surgery from Patients with Poor Left Ventricular Function

**DOI:** 10.3390/medicina58091220

**Published:** 2022-09-05

**Authors:** Roya Ostovar, Max Schmidt, Filip Schroeter, Ralf-Uwe Kuehnel, Jacqueline Rashvand, Martin Hartrumpf, Johannes Maximilian Albes

**Affiliations:** Department of Cardiovascular Surgery, Heart Center Brandenburg, University Hospital Brandenburg, Faculty of Health Sciences Brandenburg, Medical School Theodor Fontane, 16321 Bernau, Germany

**Keywords:** left ventricular ejection fraction, mitral valve surgery, mortality

## Abstract

*Background and Objectives*: Increasing reluctance to perform surgical mitral valve repair or replacement particularly in high-risk patients with poor left-ventricular function is trending. These patients are increasingly treated interventionally, e.g., by MitraClip, but often show only low to moderate improvement. The primary objective of the study was to investigate whether left ventricular ejection fraction (LVEF) influences postoperative mortality. *Materials and Methods*: The study included 903 patients undergoing mitral valve repair or replacement between 2009 and 2021. Statistical comparison was performed between patients with LVEF ≤ 30% and LVEF > 30%. Finally, statistical analysis was performed according to propensity score matching (1:3 PS matching). *Results:* No significant difference in in-hospital mortality was found before and after matching regarding LVEF ≤ 30% and LVEF > 30% (Pre: 10.8% vs. 15.1%, *p* = 0.241, after: 11.6% vs. 18.1%, *p* = 0.142). After PS matching, the 112 patients with LVEF ≤ 30% compared with 336 patients with LVEF > 30% showed a significantly higher preoperative NT-proBNP (*p* < 0.001), larger diameters at preoperative left ventricle and atrium (*p* < 0.001), lower preoperative TAPSE (*p* = 0.003) and PAP (*p* = 0.003), and more dilated cardiomyopathy and chronic kidney disease (*p* < 0.001, *p* = 0.045). *Conclusions:* The results of this study demonstrate that poor preoperative LVEF alone does not play a significant role in postoperative outcome and long-term mortality. Prognosis appears to be multifactorial. Poor preoperative LVEF is not a contraindication for surgery and does not justify primary interventional treatment accepting inferior hemodynamic results impeding outcome.

## 1. Introduction

The treatment of mitral regurgitation as one of the most common types of valvular heart disease is often complex [1]. Due to the poor outcome, conservative therapy is justifiable only in a few non-operable patients [2]. The long-term outcome of most interventional catheter-based procedures has not yet been properly investigated. The increasing use of MitraClip implantation is inferior to surgical treatment and is often considered as a last option in inoperable patients due to the unconvincing short- and long-term results. The surgical option is also not always an appropriate choice for high-risk patients. Thus, the indication for surgical mitral valve repair or replacement in high-risk patients is becoming increasingly conservative. These risks are often related to the multiple comorbidities in an ageing population. There are various risk factors that are empirically supported by clinical experience or proven by studies to make the decision for surgical treatment more challenging. These include the typical risk-factors already addressed in the EuroSCORE system and accumulating to a respective individual risk [3,4,5,6,7,8,9,10,11]. As a consequence, these patients are more and more often treated with a MitraClip, although the surgical treatment option would yield better and more stable results with the chance for the myocardium and other organ systems to recover sustainably. The primary objective of the study was to investigate how far LVEF influences postoperative mortality. The secondary purpose was to identify other risk factors influencing mortality.

## 2. Patients and Methods

### 2.1. Ethical Statement

An ethics vote was obtained from the ethics committee of Brandenburg Medical School prior to the start of data collection (E-02-20200923, dated 21 November 2020). Due to the retrospective design of the study and anonymization of the data, the necessity of informed consent was waived.

### 2.2. Data Collection

The study included 903 patients undergoing isolated mitral valve repair or replacement with or without simultaneous tricuspid valve repair between 2009 and 2021 retrospectively. Patients with infective endocarditis, combination surgery of mitral valve with coronary revascularization, aortic valve or pulmonary valve replacement, and aortic replacement, and patients < 18 years were excluded from the study.

The objective was to investigate risk factors influencing in-hospital and 30-day mortality. The main purpose was to evaluate the influence of severely reduced left ventricular ejection fraction (LVEF).

Comprehensive data were collected on baseline, risk profile, comorbidities, and type and duration of surgery.

Postoperative outcomes included complications, revision surgery, and in-hospital mortality as well as duration of hospitalization and length of stay in the intensive care unit.

The laboratory data were documented preoperatively from the day of admission and on the 7th postoperative day ± 2 days. The findings of both transthoracic and transesophageal echocardiography were used preoperatively and on the 7th postoperative days ± 2 days.

Chronic kidney failure (CKD) was classified according to the National Kidney Foundation based on five severity levels. Coronary artery disease (CAD) was considered to be present if it was described in invasive coronary angiography. According to the involvement pattern of the coronary vessels, a differentiation was made between one-, two-, and three-vessel CAD. Regarding myocardial infarction, a differentiation was performed whether it was acute or occurred in the past (>30 days).

In the case of liver disease, differentiation was also performed between steatosis hepatic, liver cirrhosis, hepatitis, etc., and in the case of lung disease between chronic obstructive pulmonary disease, pulmonary hypertension, etc.

### 2.3. Statistical Analysis

Statistical analysis of anonymized data was performed with SPSS Version (26.0, IBM, Chicago, IL, USA, 2019) and R (R Core Team, Wien, Austria) [12]. 

Categorical variables were analyzed using the Fishers exact test or the Chi Square test. Continuous parameters were initially tested for normal distribution using the Shapiro–Wilk test. In the case of borderline significance, normal distribution was assessed based on inspection of frequency histograms. Normally distributed data were compared using Student’s *t*-test, and analysis of non-normally distributed data was performed using the Mann–Whitney U-test.

The investigation of possible risk factors influencing mortality was performed by calculating the odds ratio (OR). If risk factors were statistically significant, a logistic regression analysis was subsequently performed. Here, “survival at discharge” was defined as the dependent outcome variable, and the influence of potential risk factors was examined as the independent variable. The 30-day survival and overall survival during hospitalization were descriptively presented using a Kaplan–Meier method. This was followed by a statistical comparison of both groups using the “log-rank” and “Wilcoxon test”. 

Statistical comparison was performed between patients with LVEF ≤ 30% and LVEF > 30%. A total of 130 patients had LVEF ≤ 30% and 773 patients had LVEF > 30%. 

Finally, statistical analysis was performed according to propensity score matching (1:3 PS matching). LVEF > 30% or ≤30% were used as the dependent variables and logistic EuroSCORE, age, gender, mitral valve replacement or repair, primary or secondary mitral regurgitation, and cross-clamp time were used as independent variables. A patient collective of *n* = 448 patients with a group size of *n* = 112 for LVEF ≤ 30% and *n* = 336 for LVEF > 30% was achieved. Matching was performed using nearest neighbor matching with logistic regression (Table 1). The value of LVEF alone to predict in-hospital mortality was compared with N-terminal prohormone of brain natriuretic peptide (NT-proBNP), tricuspid annular plane systolic excursion (TAPSE), pulmonary artery pressure (PAP), left ventricular diastolic diameter, logistic EuroSCORE, and age by creating receiver operating characteristic (ROC) curves using the ROCR package in R (R Core Team, Wien, Austria) [1]. Area under the curve (AUC) values were calculated as a measure of predictive quality. The significance level was defined as α = 0.05 for all tests.

### 2.4. Surgical Procedures

Mitral valve procedures were combined with tricuspid valve repair in 260 patients. In terms of mitral valve surgery, 434 mitral valve reconstructions and 469 mitral valve replacements were performed.

## 3. Results

A total of 43.5% of participants were female (*n* = 393) and 56.5% were male (*n* = 510). The mean age was 65.7 ± 12.2 years, mean logistic EuroSCORE 17.3 ± 17.3, mean EuroSCORE 8.4 ± 8, mean body mass index (BMI) 27.9 ± 5.52, and mean preoperative LVEF 51.4% ± 13.68%.

### 3.1. Baseline

A total of 131 patients died in the hospital. The non-survivors were significantly older (mean 6 years) than survivors (*p* < 0.001). The logistic EuroSCORE was significantly higher in non-survivors than in surviving patients (*p* < 0.001). History of cardiac surgery accounted for 21.9% of non-survivors and 13.5% of survivors (*p* = 0.02). NT-proBNP was higher in patients who died than in survivors (6916 ± 9871 pg/mL vs. 3385 ± 6005 pg/mL, *p* < 0.001). TAPSE was significantly lower and PAP significantly higher in non-survivors than in survivors (16.1 vs. 21.5 mm, *p* < 0.001, 53.9 vs. 33.8 mmHg, *p* < 0.001) (Table 2). 

### 3.2. Procedural Data

Of a total of 643 patients with isolated mitral valve surgery, 83 patients (12.9%) died. Simultaneous mitral valve surgery in combination with tricuspid valve repair was performed in 260 patients. Of these, 48 patients died (18.5%). The non-survivors received concomitant tricuspid repair significantly more often (36.6% vs. 27.5%, *p* = 0.041).

Patients after mitral valve reconstruction showed significantly lower mortality (7.6%, 33/434) than patients after mitral valve replacement (20.9%, 98/469) (*p* < 0.001). Mean cardiopulmonary bypass time was longer in non-survivors, 180.4 ± 75 min, compared with survivors, 168 ± 69 min, without reaching significance levels (*p* = 0.144). X-clamp time was also longer in non-survivors, 114.2 ± 47 min, compared to survivors, 109.7 ± 51 min (*p* = 0.235). In 70 patients, an intra-aortic balloon pump (IABP) was implanted perioperatively.

### 3.3. Comorbidities

The non-survivors showed significantly more arterial hypertension (84.6 vs. 74.4%, *p* = 0.02) and dilated cardiomyopathy (11.4% vs. 5.3%, *p* = 0.016). Coronary artery disease was significantly more prevalent in non-survivors than in survivors (40.7% vs. 30.4%, *p* = 0.032). By the Cochran Armitage test for trend, it could be seen that the more severe the CAD was, the higher the trend for mortality (*p* < 0.001) was. Of the non-survivors, 15.5% showed myocardial infarction in the past, while this was present in only 5.4% of survivors (*p* < 0.001). Similarly, the non-survivors showed almost twofold more CKD (34.2%) than the survivors (22.8%) (*p* = 0.01). In non-survivors, 16.4% had chronic obstructive pulmonary disease (COPD), which was more than in survivors, 10%, however, the difference was not significant. Regarding PAD, no significant difference was seen between deceased and survivors, either (Table 3). 

### 3.4. Impact of Left Ventricular Ejection Fraction on Mortality (before Propensity Score Matching)

A total of 130 patients with LVEF ≤ 30% and 773 patients with LVEF > 30% were identified. The patients with LVEF ≤ 30% compared to the others showed a significantly higher logistic EuroSCORE (16.26% vs. 14.19%, *p* = 0.032). The non-survivors with lower LVEF also showed a higher logistic EuroSCORE, 36.56%, compared to non-survivors with LVEF > 30% 31.61%, however, without reaching significance levels (*p* = 0.251).

Patients with LVEF ≤ 30% had higher preoperative NT-proBNP than the patients with LVEF > 30% (9119 vs. 2973, *p* < 0.001). Similarly, preoperative NT-proBNP was significantly higher in non-survivors with LVEF ≤ 30% than non-survivors with LVEF > 30% (14,465 vs. 5972, *p* = 0.03). 

Patients with LVEF ≤ 30% showed a significantly larger preoperative diastolic diameter of the left ventricle (58.24 vs. 52.02 mm, *p* < 0.001) and significantly lower PAP (32 vs. 38 mmHg, *p* = 0.001) than patients with LVEF > 30%. In non-survivors with LVEF ≤ 30%, a significantly larger preoperative diastolic diameter of left ventricle (58.9 vs. 49.8 mm, *p* = 0.014) and lower PAP (44.9 vs. 55 mmHg, *p* = 0.017) were also seen. Preoperative TAPSE was also significantly worse in patients with LVEF ≤ 30% than those with LVEF > 30% (18.08 vs. 20.96 mm, *p* < 0.001). The same results were reflected in non-survivors with LVEF ≤ 30% (12.29 vs. 16.55 mm, *p* = 0.004). (Table 4)

No significant difference in hospital mortality was observed between patients with LVEF ≤ 30% and >30% (10.77 vs. 15.14%, *p* = 0.24).

### 3.5. Comorbidities

It could be observed that patients with LVEF ≤ 30% had significantly more PAD than patients with better LVEF (18.7% vs. 4.02, *p* < 0.001). However, this significant difference was no longer demonstrable between non-survivors with LVEF > 30% or ≤30%. 

Dilated cardiomyopathy was also significantly more frequent in patients with LVEF ≤ 30% (8.94 vs. 3.46, *p* = 0.011). Dilated cardiomyopathy was observed significantly more often in non-survivors with LVEF ≤ 30% compared with those with LVEF > 30% (61.54 vs. 5.45, *p* < 0.001). (Table 4)

### 3.6. Postoperative Course

There was no significant difference between the patients with poor LVEF (≤30%) and those with LVEF > 30% regarding postoperative complications investigated, such as systemic inflammatory response syndrome (SIRS), renal insufficiency, apoplexy, low cardiac output syndrome, postoperative hemorrhage, pericardial effusion and tamponade, pleural effusion, critical illness polyneuropathy and myopathy, pneumonia, pneumothorax, wound healing disorder, urinary tract infection, delirium, and re-thoracotomy. However, there were significant differences in postoperative complications between survivors and non-survivors. Thus, the non-survivors showed significantly higher SIRS (*p* < 0.001), low cardiac output syndrome (*p* < 0.001), pericardial tamponade (*p* < 0.037), pneumonia (*p* = 0.038), postoperative hemorrhage (*p* = 0.043), and resulting re-thoracotomy (*p* = 0.013).

### 3.7. Impact of LVEF on Mortality (after Propensity Score Matching)

After propensity score (PS) matching, the 112 patients with LVEF ≤ 30% compared with 336 patients with LVEF > 30%, as well as and the non-survivors with LVEF ≤ 30% (*n* = 13) and LVEF > 30% (*n* = 99) showed significantly higher preoperative NT-proBNP (*p* < 0.001, 0.043), larger diameters at preoperative left ventricle and atrium (*p* < 0.001, *p* = 0.015), lower preoperative TAPSE (*p* = 0.003, *p* = 0.016) and preoperative PAP (*p* = 0.003, *p* = 0.029), more dilated cardiomyopathy (*p* < 0.001 and *p* < 0.001), and more CKD (*p* = 0.045, *p* = 0.029). However, no significant difference in mortality was found (*p* = 0.142). (Table 5).

Figure 1 shows the ROC curves and AUC values of LVEF, NT-proBNP, TAPSE, PAP, logistic EuroSCORE, diastolic diameter of the left ventricle, and age for the prediction of in-hospital mortality. While PAP, logistic EuroSCORE, and TAPSE have a good predictive value, as expected, the comparison shows that LVEF and diastolic diameter of the left ventricle alone have the lowest and thus a rather poor predictive value surpassed even by age alone.

## 4. Discussion

The results show that many parameters influenced survival in our cohort. These include, as expected, high logistic EuroSCORE and those factors that contribute to the calculation of EuroSCORE, such as redo-surgery, age, history of myocardial infarction, and elevated PAP with a mean value of 50 mmHg in non-survivors as an expression of advanced pulmonary hypertension. However, additional parameters with a direct impact on mortality were also found. These included coronary artery disease, CKD, presence of dilated cardiomyopathy, poor right ventricular function as determined by TAPSE, or elevated NT-proBNT as an index of advanced heart failure. 

Perhaps of interest, concomitant tricuspid repair does not always seem to be unequivocally beneficial and thus justified as mortality increased with the degree of intraoperative stress such as additional tricuspid repair. The negative influence of prolonged clamp time on survival, however, did not reach a significant level. Very well in line with the current evidence, the positive effect of mitral valve repair as compared to replacement regarding survival was shown. 

To our surprise, we did not find significant differences in mortality after comparing patients with significantly reduced LVEF (≤30%) and patients with preserved LVEF (>30%). However, there were a large number of statistically relevant differences between patients with poor and better LVEF. Patients with poor LVEF (≤30%) had significantly more accompanying extracardiac diseases, but also worse cardiac conditions, such as higher NT-proBNP, enlarged left ventricular diameter, or lower TAPSE. Surgical factors such as concomitant tricuspid repair also played a role. Even after propensity score matching, patients with poor LVEF showed similar results to those with better LVEF. 

Reduced LVEF alone does not appear to have a direct effect on mortality. One aspect may simply be that LVEF alone is not a reliable parameter of myocardial function in patients with mitral regurgitation [13,14]. Regurgitation makes it almost impossible to accurately assess the remaining myocardial capacity after correction of valve function. Therefore, it can be assumed that patients with poor and with apparently preserved function belong to the same cohort and behave quite similarly. In contrast, it seems that the overall cardiopulmonary condition and concomitant diseases in combination as addressed by the EuroSCORE play an important role in mortality. Particularly, the state of the right ventricle and pulmonary vasculature utilizing PAP and TAPSE can serve as predictive parameters for an adverse outcome.

## 5. Limitation

Owing to the retrospective nature of the study, many parameters were missing. Furthermore, a variety of parameters were not as reliable as desired. This is especially relevant for echocardiography with its high investigator-related inconsistencies. Another limitation is that, because of the small number of patients with LVEF ≤ 30% (N = 112/903), propensity score matching had to be limited to a certain number of parameters in order to maintain statistical power. Thus, a more elaborate, in-depth comparison was not possible. 

## 6. Conclusions

Left ventricular ejection fraction may play a role in mortality, but most likely does not have the status which is often given to it. Thus, patients should not be denied surgery in a heart team mainly based upon low LVEF, but should rather be evaluated in a thorough, holistic fashion in order to find the most appropriate individual approach. Special attention should be paid to the condition of the right ventricle and pulmonary vessels. Furthermore, in selecting patients for tricuspid valve repair, the patient’s overall condition should be considered. This, as already mentioned, includes simultaneous presence of multiple risk factors such as pulmonary hypertension, poor right and left ventricular function, concomitant disease, and extent of NT-proBNP.

Moreover, mitral valve repair should be performed whenever possible. Patients after repair show better outcomes in terms of hospital survival compared to replacement. The repair does offer not only hemodynamic advantages by preserving the subvalvular apparatus but also other long-term benefits such as avoidance of mandatory anticoagulation or at least a more generous preference for new oral anticoagulants instead of phenprocoumon. Valve replacement is much riskier than repair because it can have serious consequences, such as atrioventricular rupture, paravalvular leakage requiring immediate reoperation with prolonged ischemia time, left ventricular outflow tract problems or development of endocarditis in the further course. Replacement should thus be reserved only to patients who have no chance at all of an anatomically successful and sustainable repair. 

It should be mentioned that the results and conclusions refer to the study patients and may be different in another group or group size.

## Figures and Tables

**Figure 1 medicina-58-01220-f001:**
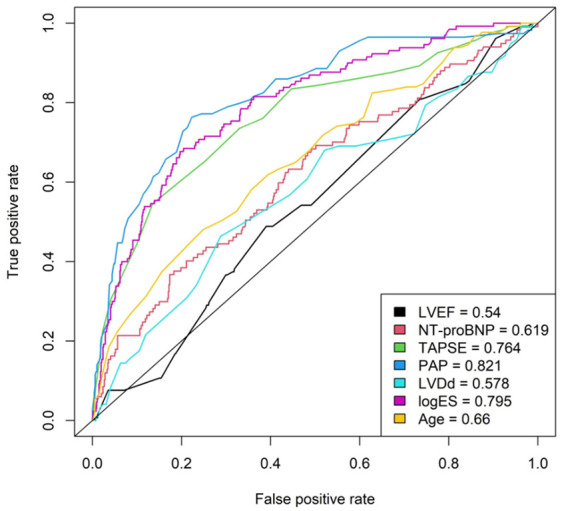
AUC: area under curve, EF: left ventricular ejection fraction, NT-proBNP: N-terminal prohormone of brain natriuretic peptide. TAPSE: tricuspid annular plane systolic excursion, PAP: pulmonary artery pressure, LVDd: diastolic diameter of left ventricle, logES: logistic EuroSCORE.

**Table 1 medicina-58-01220-t001:** Propensity score matching.

Before PS Matching	LVEF ≤ 30% (*n* = 130)	LVEF > 30% (*n* = 773)	*p*-Value
Age	66.65 ± 11.49	65.72 ± 12.36	0.637
Logistic EuroSCORE	16.62 ± 17.67	14.19 ± 18.18	0.032
Gender (female)	36.92% ±48	44.63% ± 345	0.122
Secondary mitral regurgitation	20.87% (24)	7.66% (36)	<0.001
Proportion of mitral valve replacement	53.08% (69)	47.22% (365)	0.253
Cross-clamp time (min)	109.2 (47.38)	110.6% (51.5)	0.783
**After PS matching**	**LVEF ≤ 30%** **(*n* = 112)**	**LVEF > 30%** **(*n* = 336)**	***p*-Value**
Age	65.96 ± 11.57	66.07 ± 12.31	0.782
Logistic EuroSCORE	16.84 ± 18.45	15.72 ± 20.36	0.093
Gender (female)	35.71% ± 40	35.1% ± 120	1
Secondary mitral regurgitation	19.64% (22)	10.71% (36)	0.023
Proportion of mitral valve replacement	52.68% (59)	50.30% (169)	0.743
Cross-clamp time (min)	109.02 ± 47.34	109.41 ± 43.67	0.796

PS matching: propensity score matching, LVEF: left ventricular ejection fraction.

**Table 2 medicina-58-01220-t002:** Baseline and surgical data.

Baseline	Non-Survivors *N* = 131	Survivors *N* = 772	*p*-Value
Age (years)	71.2 ± 9.9	64.8 ± 12.4	<0.001
Logistics EuroSCORE (%)	32.1 ± 24.9	11.5 ± 14.7	<0.001
Body mass index (Kg/m^2^)	29.1 ± 7	27.7 ± 5	n.s.
Cardiac surgery in history	21.9%	13.5%	0.02
Preoperative NT-proBNP (pg/mL)	6916 ± 9871	3385 ± 6005	<0.001
Preoperative LVEF (%)	50.1%	51.68%	0.136
Preoperative TAPSE (mm)	16.1 ± 5.7	21.5 ± 5.8	<0.001
Preoperative PAP (mmHg)	53.9 ± 16.8	33.8 ± 13.6	<0.001
Mitral valve repair vs. replacement	25.2% vs. 74.8%	51.9% vs. 48.1%	<0.001
Proportion of tricuspid repair	36.6 ± 48	27.5 ± 212	0.041
Clamp time (min)	114 ± 48	109 ± 51	0.235
intra-aortic balloon pump	7.7%	7.7%	1

NT-proBNP: N-terminal prohormone of brain natriuretic peptide, LVEF: left ventricular ejection fraction, TAPSE: tricuspid annular plane systolic excursion, PAP: pulmonary artery pressure.

**Table 3 medicina-58-01220-t003:** Influence of comorbidities on mortality.

Comorbidities	Non-Survivors *N* = 131	Survivors *N* = 772	*p*-Value
Arterial hypertension	84.6%	74.4%	0.02
Peripheral artery disease	5.7%	4.0%	0.543
Myocardial infarction	15.5%	5.4%	<0.001
Dilated Cardiomyopathy	11.4%	5.3%	0.016
Coronary artery disease - 1 vessel - 2 vessels - 3 vessels	40.7% 9.8% 6.5% 23.6%	30.4% 13.1% 7.9% 9.5%	0.032
Chronic obstructive pulmonary disease	16.4%	10.0%	0.543
Chronic kidney disease - Stage I - Stage II - Stage III - Stage IV - Stage V	34.2% 8.1% 1.6% 16.3% 4.1% 4.1%	22.8% 10.1% 3.2% 8.2% 0.8% 0.7%	0.01

**Table 4 medicina-58-01220-t004:** Risk profile and relation to mortality in patients with LVEF ≤ 30% and >30% before propensity score matching.

	All Patients	*N* = 903		Non-Survivors	*N* = 131	
	LVEF ≤ 30% *N* = 130	LVEF > 30% *N* = 773	*p*-Value	LVEF ≤ 30% *N* = 14	LVEF > 30% *N* = 117	*p*-Value
Logistic EuroSCORE	16.26%	14.19%	0.032	36.56%	31.61%	0.251
NT-proBNP preop	9119	2973	<0.001	14,465	5979	0.03
LVDd (mm) preop	58.24	52.02	<0.001	58.82	49.77	0.014
LADd (mm) preop	46.9	45.6	0.112	43.3	46.2	0.248
TAPSE (mm) preop	18.08	20.96	<0.001	12.29	16.55	0.004
PAP (mmHg) preop	32.01	38.33	0.001	44.92	55	0.017
Coronary artery disease	38.21%	30.84%	0.13	15.38%	43.64%	0.072
Redo surgery	16.3%	14.4%	0.694	23.1%	21.8%	1
IABP	10.9%	7.2%	0.194	14.3%	6.9%	0.293
COPD	10.1%	11.1%	0.858	0%	18.35%	0.124
Myocardial infarction	8.94%	6.51%	0.427	7.69%	16.36%	0.689
Dilated cardiomyopathy	18.7%	4.02%	<0.001	61.5%	5.45%	<0.001
PAD	8.9%	3.4%	0.011	7.7%	5.5%	0.552
CKD	30.9%	23.4%	0.093	46.2%	32.7%	0.512
Liver disease	93.5%	92.9%	0.813	69.2%	90%	0.019
In hospital Mortality	10.77%	15.14%	0.241	-	-	-

PS: propensity score matching, LVEF: left ventricular ejection fraction, NT-proBNP: N-terminal prohormone of brain natriuretic peptide. Preop: Preoperative, Postop: postoperative, LVDd: left ventricle diameter diastolic, LADd: left atrium diameter diastolic, TAPSE: tricuspid annular plane systolic excursion, PAP: pulmonary artery pressure, COPD: chronic obstructive pulmonary disease, CKD: chronic kidney disease, IABP: intra-aortic balloon pump, PAD: peripheral artery disease.

**Table 5 medicina-58-01220-t005:** Risk profile and relation to mortality in patients with LVEF ≤ 30% and >30% after propensity score matching.

	All Patients			Non-Survivors		
After PS Matching	LVEF ≤ 30% *N* = 112	LVEF > 30% *N* = 336	*p*-Value	LVEF ≤ 30% *N* = 13	LVEF > 30% *N* = 99	*p*-Value
NT-proBNP preop	8293	3041	<0.001	15,150	5771	0.043
LVDd (mm) preop	58.98	52.57	<0.001	59.7	50.02	0.015
LADd (mm) preop	47.47	45.94	0.147	43.31	45.52	0.468
TAPSE (mm) preop	17.87	20.3	0.003	11.92	15.41	0.016
PAP (mmHg) preop	32.5	39.3	0.003	44.6	55.1	0.029
Coronary artery disease	36.4%	36.8%	1	15.4%	42.6%	0.113
Dilated cardiomyopathy	20%	4.79%	<0.001	61.5%	4.9%	<0.001
COPD	10.38%	11.62%	0.861	0%	15%	0.347
CKD	66.4%	75.7%	0.045	53.8	78.7	0.029
Need for IABP	11.7%	7.49%	0.236	15.4%	8.2%	0.599
In-hospital Mortality	11.6%	18.2%	0.142	-	-	-

PS: propensity score matching, LVEF: left ventricular ejection fraction, NT-proBNP: N-terminal prohormone of brain natriuretic peptide. Preop: Preoperative, LVDd: left ventricle diameter diastolic, LADd: left atrium diameter diastolic, TAPSE: tricuspid annular plane systolic excursion, PAP: pulmonary artery pressure, COPD: Chronic obstructive pulmonary disease, IABP: intra-aortic balloon pump, CKD: chronical kidney disease.

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
