# Peer review of "Do Not Withhold Mitral Surgery from Patients with Poor Left Ventricular Function"

_medicina, 2022, doi:10.3390/medicina58091220_

Round 1
Reviewer 1 Report
In the present study authors seek to investigate influence of LVEF in postoperative mortality in patients undergoing mitral valve surgery (repair or replacement) by applying propensity score matching.
In the introduction references pointed by the authors did not reflect real actual practice. Realistic and solid registries as the German registry with 628 patients with procedural failure reported as 3.2% and mortality is reported 20.3% with 30.8% patients with LVEF< 30%. Moreover, 63.3% patients were reported to be at NYHA I-II at 1 year.
Regarding the methods, the authors decide to perform propensity score matching using LVEF, nonetheless etiology (functional or degenerative) and medical treatment are not included in the propensity score. I do not know if this fact would have changed the results but the authors need to explain why they did it. The authors need to bear in mind the fact that etiology matters in mitral valve surgery.
The conclusion made by the authors are too general, they could be improved, sometimes conclusions are the first paragraph we read.
Author Response
Dear Reviewer,
We would like to thank you for careful and thorough reading of this manuscript and for the thoughtful comments. The suggested correction has been made. So we have sent the revised manuscript, and a version containing all the changes to be visible.
Comment 1: In the introduction references pointed by the authors did not reflect real actual practice. Realistic and solid registries as the German registry with 628 patients with procedural failure reported as 3.2% and mortality is reported 20.3% with 30.8% patients with LVEF< 30%. Moreover, 63.3% patients were reported to be at NYHA I-II at 1 year.
Answer 1: We are not entirely certain which registry is meant. Perhaps, the Registry of the German Society of Thoracic-, Cardiac-, and Vascular Surgery (DGTHG). The annual report of 2021 which is already visible online on the DGTHG Website (see new reference) but not yet published refers to 6052 mitral valve procedures. Early mortality is not specified. Specified, however, is the mortality of all valve procedures with 3.3%.
changes: reference 13
Comment 2: Regarding the methods, the authors decide to perform propensity score matching using LVEF, nonetheless etiology (functional or degenerative) and medical treatment are not included in the propensity score. I do not know if this fact would have changed the results but the authors need to explain why they did it. The authors need to bear in mind the fact that etiology matters in mitral valve surgery.
Answer 2: Thank you for the valuable comment. On request of reviewer 2, we changed the study population to mitral valve replacement and repair with or without tricuspid repair only. In the new analysis, both primary and secondary mitral regurgitation and type of treatment repair vs valve replacement were considered in propensity score matching.
Changes: Page 3 and table 1
Comment 3: The conclusion made by the authors are too general, they could be improved, sometimes conclusions are the first paragraph we read.
Answer 3: We agree. The conlusion was supplemented
Changes: pages 8 and 9.
Reviewer 2 Report
Dear Authors,
Thank you for you manuscript.
Comments:
1. You presented that 1301 pts underwent MV surgery. On the other hand 827 pts underwent combined operations (309 with CABG, 307 with HV AND 181 with CAGB). Only 474 pts underwent isolated MV surgery. I think that your study population is completely heterogenic and combined operation affect directly your result, even you done PS matching. Please revise your study population and include the pts who underwent MV surgery only or MV surgery+TV surgery. Please exclude pts with MV+CABG and MV+AV surgery
2. Please provide NT-proBNP units in the text.
Thank you
Author Response
Dear Reviewer,
We would like to thank you for careful and thorough reading of this manuscript and for the thoughtful comments. The suggested correction has been made. So we have sent the revised manuscript, and a version containing all the changes to be visible.
Comment 1: You presented that 1301 pts underwent MV surgery. On the other hand 827 pts underwent combined operations (309 with CABG, 307 with HV AND 181 with CAGB). Only 474 pts underwent isolated MV surgery. I think that your study population is completely heterogenic and combined operation affect directly your result, even you done PS matching. Please revise your study population and include the pts who underwent MV surgery only or MV surgery+TV surgery. Please exclude pts with MV+CABG and MV+AV surgery.
Answer 1: we have now excluded all combination procedures from the study except for tricuspid valve repair. In order not to run out of patients, we have extended the data collection until 2021, as planned earlier. The results are of course different in numbers, but main message remained identical. Interestingly, in new results (isolated consideration of only mitral valves with or without tricuspid surgery) pulmalary artery pressur is relevant regarding mortality. So thank you for your comment.
Changes: Throughout manuscript marked in yellow.
Comment 2: Please provide NT-proBNP units in the text.
Answer 2: Thank you very much. The unit was added.
Changes: page 4 to 6, table 2 and 4
Round 2
Reviewer 1 Report
There is a mistake in Table 1 regarding group sizes.
The authors may clarify what is the definition accepted for cardiomyopathy, does it refer to dilated cardiomyopathy?
In page 8, line 274 and 275 authors say that patients with reduced LVEF represent a small population, only 82 patients, at baseline they refer to 130 patients with LVEF ≦30% and after PS matching they report 112 patients, finally analyzing 80 patients.
Authors should emphasize that the conclusions made by them only apply to their population and should not be generalized.
Author Response
Comment 1: Rev. There is a mistake in Table 1 regarding group sizes.
Answer 1 : thank you very much. We made the mistake after 1st review and have corrected it now.
Change 1: Table 1, page 3
Comment 2: The authors may clarify what is the definition accepted for cardiomyopathy, does it refer to dilated cardiomyopathy?
Answer 2: You are absolutely right. dialatile cardiomyopathy was added accordingly.
Change 2: page 1, 4, 5, 6 und 8
Comment 3: In page 8, line 274 and 275 authors say that patients with reduced LVEF represent a small population, only 82 patients, at baseline they refer to 130 patients with LVEF ≦30% and after PS matching they report 112 patients, finally analyzing 80 patients.
Answer 3: we understand that it seems a bit confusing.
In the cohort were a total of 903 patients. There were 3 main statistical analyses performed independently of each other with partial subanalyses.
1st analysis: before PS matching, there were 130 patients with EF <30% and 773 patients with EF >30%.
2nd analysis: after PS matching, there were 112 patients with EF <30% and 336 patients with EF >30%.
3rd analysis: in total cohort (903 patients) were 131 non-survivors and 772 patients.
In addition, some subanalyses & descriptive descriptions e.g. regarding mortality were performed. Originally (before the 1st review), however, we had 82 patients with EF <30% after PS matching. Unfortunately, during the review, we forgot to correct this number in Table 1 and page 8. Fortunately, you noticed. this has now been corrected
Change 3: Table 1 and page 8
Comment 4: Authors should emphasize that the conclusions made by them only apply to their population and should not be generalized.
Answer 4: We agree. we have added an additional sentence
change 4: conclusion, page 9
Reviewer 2 Report
Dear Authors,
Thank you.
You answer in most reviewers questions.
Please avoid self-citations (ref 3 , 4)
Thank you
Author Response
Comment 1: Please avoid self-citations (ref 3 , 4)
Answer 1: Thank You. We agree. Change was made